# Limited Intervention and Moral Kindergartens

## Daniel Lim

Arts & Humanities Department, Duke Kunshan University, Kunshan 215316, China; daniel.lim672@duke.edu

**Abstract:** Recently, William Hasker and Cheryl Chen have argued that James Sterba's argument for the non-existence of God based on the existence of horrendous evil consequences fails. Hasker, among other things, contends that eliminating horrendous evil consequences will result in a moral kindergarten. It is unclear, however, whether the elimination of horrendous evil consequences will result in a moral kindergarten. Moreover, if Hasker is right, then it may be that most people in the actual world live in a moral kindergarten. Chen argues that eliminating horrendous evil consequences may lead to a morally worse world. While Chen is ultimately right about this, it is not fatal to the basic intuition at the heart of Sterba's argument.

**Keywords:** problem of evil; moral kindergarten; James Sterba; William Hasker; Cheryl Chen

## 1. Introduction

In *Is a Good God Logically Possible?* James Sterba develops and defends an argument against the existence of God based on the existence of significant and especially horrendous evil consequences of immoral actions. According to Sterba, if God exists, then God must govern the world according to certain basic moral requirements because God is good. For example, consider the Moral Evil Prevention Requirement I (MEPR1):

> Prevent, rather than permit, significant and especially horrendous evil consequences of immoral actions without violating anyone's rights (a good to which we have a right) when that can easily be done. (Sterba 2019, p. 126)

Assuming MEPR1 is exceptionless, Sterba argues that the existence of horrendous evil consequences of immoral actions that God could have easily prevented without violating anyone's rights is logically incompatible with the existence of God. Since such horrendous evil exists, it follows that God does not exist. The remainder of this paper will be devoted to an examination of two recent, independent attacks on MEPR1.

## 2. Hasker–Sterba Debate

Hasker (2004, 2020, 2021) rejects MEPR1. His rationale, briefly, is that human soul-making takes precedence over MEPR1 and thereby creates exceptions for MEPR1. In his 2004 treatment of this issue, Hasker discusses the tension between God's intention for soul-making (which he calls the divine moral imperative or DMI) and MEPR1 (which he calls no gratuitous evil or NGE). He writes:

> It seems evident to me that DMI [or soul-making] is far more deeply entrenched in the theistic worldview than is NGE [or MEPR1], so that the tension between them is an indication that NGE [or MEPR1] should be abandoned. (Hasker 2004, p. 89)

The idea is that in a world where God acts according to MEPR1, among other things, humans would (i) not have the kind of free will that is necessary for making significant moral decisions or (ii) not have sufficient motivation to act against evil since evil actions would never result in horrendous consequences. Humans in such a world would be living in a 'moral kindergarten', where God allows us to 'argue over blocks', but never lets anyone actually get hurt. Let's call this the Moral Kindergarten Response (MKR). Hasker is not

alone in defending the MKR. Here is a sampling of two 'standard' contemporary variations on this response:

> But God could of course arrange things so that our bad choices never had any effects. When we chose kind words, they came out of our mouth; when we chose to insult, the air did not convey the message. When we chose to strike, we became paralysed; when we chose to stroke, our hands obeyed our commands. But for God to create agents permanently so placed would be a great deceit. He would have made it seem to us as though we had power—for we could not have a choice between uttering kind words and uttering insulting words, unless we thought our attempts to talk would be successful. (Swinburne 1998, pp. 144–45)

> God should put us all in a virtual playpen in which choices can be made without any real harm to others being caused. Good choices could be made, and the good consequences that follow from them allowed. But bad choices, while not prevented altogether, would be prevented from causing additional damage. Couldn't God simply block such negative outcomes? . . . [This would come] at the price of keeping us from being able to make genuinely morally significant choices between good and evil alternatives. (Murray and Rea 2008, p. 173)

Sterba's formulation of MEPR1, however, anticipates the 'standard' MKR. He agrees that eliminating all evil consequences from immoral actions will undermine human soul-making. A more nuanced way of dealing with evil is to eliminate only the 'horrendous' evil consequences while preserving the possibility of non-horrendous evil consequences. Let us call a world in which God regularly intervenes in this way, a HORRENDOUS-less world.

Here's how a scenario in a HORRENDOUS-less world might pan out. A child is being abducted. You have the ability to prevent the abduction. You choose, however, not to intervene. God also (at least at the moment) chooses not to intervene. Consequently, the abductors successfully drive off with the child. Only after the child is taken does God intervene. Perhaps God makes it so that the taillight of the abductors' car fails, and the abductors are stopped by a passing patrol car. This eventually leads to the freeing of the child. Though the child is physically unharmed and spared from horrendous evil consequences, the child is nevertheless psychologically traumatized.

Sterba summarizes what this HORRENDOUS-less world would look like at a more abstract level:

> When you choose to intervene to prevent horrendously evil consequences, either you will be completely successful in preventing those consequences or your intervention will fall short. When the latter is going to happen, God does something to make the prevention completely successful. Likewise, when you choose not to intervene to prevent such consequences, God again intervenes but not in a way that is fully successful. Here, there is a residue of evil consequences that the victim still does suffer. This residue is not a horrendous evil but it is a significant one, and it is something for which you are primarily responsible . . . [and leaves] you with an ample opportunity for soul-making. (Sterba 2021, p. 2)

Though horrendous evil consequences would be absent, the residue of evil consequences (hereafter, simply residue) would exist and make human soul-making possible. Thus, the MKR, at least when developed in the 'standard' way similar to Swinburne (1998) and Murray and Rea (2008), fails to address Sterba's limited intervention response.

Hasker, however, is fully aware of these nuances. Nevertheless, he asserts that Sterba's position remains vulnerable to the MKR:

> But if all the significant evil consequences of all immoral actions were thus prevented, agents would surely become aware that actions that would seriously harm other persons would fail to accomplish their ends; exercise of that sort of free choice would then become impossible. To be sure, some exercise of free will, even in immoral actions, would still occur, but only on relatively trivial matters. I once described this as a situation in which God was in effect running a moral

> kindergarten, allowing us to develop our characters by arguing over the blocks, but ready to intervene before anyone actually gets hurt! (Hasker 2020, p. 210)

It seems that the Sterba–Hasker debate, at least at this point, has reached an impasse. Both agree that a HORRENDOUS-less world would be absent of horrendous evil consequences while leaving room for the residue. They disagree, however, about what the residue makes possible. Sterba believes that the residue allows for robust human soul-making. Hasker believes that the residue does not allow for robust human soul-making.

Though it seems Sterba and Hasker have dug in their heels, I believe we can make modest progress in at least two ways. First, a little care with the terms and descriptions used to refer to the differing levels of severity of evil consequences may help. Sterba and Hasker use terms such as 'trivial', 'significant', and 'horrendous', but they are not always consistent with each other and with themselves. Compare, for example, how Sterba phrases his response to the MKR in 2019:

> When you choose to intervene to prevent *significantly* evil consequences, either you will be completely successful in preventing those consequences or your intervention will fall short. When the latter is going to happen, God does something to make the prevention completely successful. Likewise, when you choose not to intervene to prevent such consequences, God again intervenes but not in a way that is fully successful. Here there is a residue of evil consequences that the victim still does suffer. This residue is *not a significant* evil in its own right, but it is harmful nonetheless, and it is something for which you are primarily responsible . . . [and leaves] you with a *limited opportunity* for soul-making. (Sterba 2019, pp. 61–62, my emphasis)

Note that this passage and the one above from 2021 are nearly identical. The critical difference is that Sterba, in 2019, exclusively uses the term 'significant' to describe the evil consequences that God prevents while Sterba, in 2021, uses the term 'horrendous'. This is strategic because the kinds of soul-making opportunities that Sterba envisions as being possible are consequently different. Sterba, in 2019, says that the elimination of 'significant' evil allows for a 'limited opportunity' for soul-making, while Sterba in 2021 says that the elimination of 'horrendous' evil leaves 'significant' evil intact, and thereby, allows for an 'ample opportunity' for soul-making.

To move past these terminological differences, let us dispense with morally charged words such as 'trivial', 'significant' and 'horrendous', and opt for numbers to represent the two levels of consequences that are relevant to the present debate. Moreover, instead of defining these levels in terms of their severity, let us define them in relation to the possibility of soul-making.

Level 1: the kind of evil consequences that does not allow for soul-making.
Level 2: the kind of evil consequences that allows for soul-making.

We are now in a position to classify some of the examples of evil consequences that have been used in this debate. The pain resulting from having one's foot stepped on is level 1. The physical abuse and torture resulting from a child abduction is level 2.

One may worry, however, that this way of categorizing evil consequences is unrealistic since it ignores the subjective experience of these consequences.[1] The same consequence, after all, may be experienced in very different ways by different people. Consider two examples. A trauma victim may experience an event as deeply troubling, while another person may experience the same event as inconsequential; or consider a person who is devastated by the loss of a pet, while another person does not really feel much by the loss of a pet. If we combine the differences in the way events are subjectively experienced for different people with the claim that soul-making depends on how events are subjectively experienced, then it seems problematic to use my classification. If an evil consequence may allow for soul-making for some people but not for others, it would be unclear how this evil consequence should be classified.

The subject relativity of how consequences are experienced is equally worrisome for categorizing consequences with terms such as 'trivial', 'significant', and 'horrendous'.

After all, one person may experience an event as 'horrendous', while another person may experience the same event as 'trivial'.

Perhaps, an easy fix is to relativize levels 1 and 2 to each person. What matters then, is that there are evil consequences that are level 2 for each person in a HORRENDOUS-less world. This makes it impossible to talk about any particular consequence as being level 1 or 2 *simpliciter*. Because the majority of humans experience the various consequences discussed here (e.g., child abduction, slavery) in more or less the same way regarding soul-making, I will set aside the person relativity of levels 1 and 2 for the sake of keeping the discussion less cumbersome.

Given the level 1 and 2 categories, what are we to make of the residue? For example, what are we to make of the psychological trauma resulting from a child abduction? Should it be classified as level 1 or 2? Care is needed here. Hasker rightly notes that the severity of the psychological trauma a child experiences is affected by God's adherence to MEPR1. He writes:

> It is clear that the 'residue of evil consequences' left in cases of the sort described by Sterba would be very much less severe than what would occur without the proposed divine intervention. (Hasker 2021, p. 4)

In a world where child abductions never end in physical abuse or torture, it's difficult to predict what the abducted child would psychologically experience. The child would have no concerns over potential physical abuse or torture so how serious could the child's experience of the abduction be? Should we still classify the child's experience as 'traumatic'? What we can be sure of is that the severity of the psychological toll an abducted child undergoes would be diminished (perhaps greatly) if God adhered to MEPR1. At a minimum, this shows that the psychological experience of being abducted can no longer serve as an obvious exemplar for level 2 consequences. Indeed, something similar can be said about the other examples (e.g., providing food and shelter for a destitute person (p. 90), saving West Africans from slave trading (p. 131)) Sterba offers in his book. This then serves as a challenge for Sterba to come up with better examples—examples where the residue is (i) clearly level 2 and (ii) not affected by God's adherence to MEPR1.

A second way of making progress in the Hasker–Sterba debate is to take a closer look at the kind of soul-making that most of us *actually* go through. We seldom (if ever) have the opportunity to stop a child abduction or free slaves. The possibility of abducting a child or enslaving others is a psychological impossibility for the vast majority of us. Most of our lives happen in the banal domain of 'everyday life'. Nevertheless, I would like to argue that we are provided with 'ample opportunity' for soul-making. Consider the recent events in the U.S. and the world that are disrupting our social order: the COVID-19 pandemic, the investigation of the January 6 attack, and the overturning of Roe vs. Wade. These are testing our collective ability to adjust, develop tolerance, and live peacefully together despite radical differences in opinions and values. Even with our civil liberties protected, these are extremely challenging times. Many have been pushed to the brink physically and psychologically.

To take a personal example, my wife and I have struggled with infertility for more than 13 years. It's hard to describe the kind of difficulties we've endured during this time. I've lost count of the number of baby showers my wife has had to attend over the years and the number of times she's cried on Mother's Day while trying to be happy for all her friends with children. The remarkable ways she has persevered and learned to celebrate others despite her own inability to conceive are quite remarkable. I often look to her experiences and growth over the past years as a symbol of courage and strength that I aspire to. I realize that this is a biographical anecdote and there is nothing academically rigorous about the point I am making. However, if my wife's life were to end now, on Hasker's view, would she have had an opportunity for robust soul-making? It seems the answer would be 'no'.

The point is many people in the actual world live out their entire lives without having to directly engage the horrendous evil consequences of immoral actions. They, by and large, live only in the residue. Given this, their lives in a HORRENDOUS-less world would be no different from what their lives are like in the actual world.

One may worry that our *awareness* of the existence of horrendous evils allows for soul-making that would not be possible if such evils did not exist (See Note 1). It is unclear, however, that this awareness plays any role in the soul-making that, for example, my wife has undergone through her years of infertility. The fact that there have been such evils (e.g., slavery, genocide) in human history is something she was not consciously aware of during her struggles. Moreover, even if she were consciously aware of such evils, this awareness, if anything, might have helped to alleviate the pain she experienced by 'putting things in perspective'. Thus, the non-existence of such evils, by removing a possible means of easing her pain, might have provided a more robust soul-making opportunity for her.

This suggests a dilemma for Hasker. Many people in the actual world who live only in the residue either have or do not have the opportunity for soul-making. If many people in the actual world who live only in the residue have the opportunity for soul-making, then so do the people in a HORRENDOUS-less world. If it is impossible for people in the actual world who live only in the residue to engage in soul-making, then many of us (probably most of the people reading this paper) will never engage in soul-making. It seems to me that both are dissatisfying options for Hasker.

### 3. Chen–Sterba Debate

Chen (2021), similar to Hasker, rejects MEPR1. Her rationale, however, is different. According to Chen, it is logically possible that more people would choose to act wrongly in a HORRENDOUS-less world than the actual world. It does not matter that this is (possibly for many) unlikely and implausible. Why, after all, would the elimination of horrendous evil consequences result in more people choosing to act wrongly? Though an explanation may be desired, no explanation is needed, since we are dealing with sheer logical possibilities. If we treat logical possibilities as the absence of conceptual contradictions, Chen can freely help herself to this possibility. After all, logical possibilities of this sort are cheap. In fact, there is no need to focus only on a HORRENDOUS-less world. Since we're dealing with logical possibilities, we could just as easily work with all worlds that differ from the actual world in any way in terms of the way God intervenes to prevent evil consequences (ones where God never intervenes to ones where God always intervenes). We could confidently assert that in all these worlds, more people *may* choose to act wrongly than the actual world. This is a logical possibility. What matters is that this is logically possible and that is all that is needed to potentially undermine Sterba's argument, since his argument is cast as a *logical* argument for the non-existence of God.

Sterba grants this logical possibility. The debate between Chen and Sterba rests on the moral evaluation of a HORRENDOUS-less world where more people choose to act wrongly. Chen argues that such a world is morally worse than the actual world, while Sterba argues that such a world is *not* morally worse than the actual world. However, is it obvious that a HORRENDOUS-less world with more evil intentions is morally worse than the actual world?

This is difficult to assess because evil intentions do not always carry more (or less) moral weight than their evil consequences. Our intuitions are pulled in different directions depending on the case. On one hand, one could follow consequentialist intuitions and argue that evil consequences are morally worse than evil intentions (Mill [1861] 1998). A world where someone merely desires to hurt another but does not would be morally better than a world where someone simply gets hurt in the absence of any ill intentions. On the other hand, one could follow Kantian intuitions and argue that evil intentions are morally worse than evil consequences (Kant [1785] 1997). The morality of an action should be based solely on what is under one's control (i.e., intentions)—the consequences of such intentions are bound up in luck and are neither good nor evil. Kant writes: "Even if . . . this [intention] should wholly lack the capacity to carry out its purpose—if with its greatest efforts it should yet achieve nothing and only the good [intention] were left . . . —then, like a jewel, it would still shine by itself, as something that has its full worth in itself. Usefulness or fruitlessness can neither add anything to this worth nor take anything away from it." (Kant [1785] 1997, p. 8). The point is there is no consensus on how to assess the relative moral weight of evil

intentions and evil consequences. This complicates the moral calculus and is one of the foundational reasons we continue to have lively debates in moral philosophy.

One way to circumvent these complications is to imagine a HORRENDOUS-less world where *all* people have horrendous evil intentions *all* the time. Unlike the previous discussion where there is merely a relative increase in the number of evil intentions vs. the number of good intentions, this way of imagining the HORRENDOUS-less world results in a world that is completely saturated with evil intentions—there is not a single good intention left in this world. If this were the case, Sterba *prima facie* agrees that a HORRENDOUS-less world would be morally worse.

> Surely [a HORRENDOUS-less] world [where everyone, not just more people, would all attempt to act horrendously wrong] would be morally worse than our world, and that possibility is all that is needed to undercut my argument. (Sterba 2021, p. 9)

Upon closer inspection, however, Sterba notes that the 'inner moralities' of the people in a HORRENDOUS-less world would be equivalent to the inner moralities of the people in the actual world. We might say that they have equivalent moral dispositions. Given this equivalence, he argues that a HORRENDOUS-less world is morally better since, all else being equal, there are no horrendous evil consequences. All else, however, is not equal. There are differences in the quantity of evil intentions across these worlds, but Sterba treats these differences as superficial—mere differences in environments (not differences in inner moralities). To use an analogy, just because salt is solid in a dry environment and dissolves in a liquid environment, it does not mean that salt itself differs in these two cases. Salt is dispositionally equivalent across differing environments. However, the dispositional equivalence across environments means little when one wants to, say, package salt in tissue paper. In this case, only solid forms of salt will do. Salt dissolved in liquid is useless for this purpose. Though salt is dispositionally equivalent across these environments, it matters whether the salt is solid or dissolved. Similarly, the dispositional moral equivalence of people across worlds means little when assessing the relative moral goodness of these worlds. It matters whether evil intentions are actualized or not.

All else being equal, a world with evil intentions is morally worse than a world without evil intentions; so even if the people across different possible worlds have equivalent moral dispositions, it matters whether evil intentions exist or not. What does this tell us about the HORRENDOUS-less world we are considering? In a world where all people have evil intentions all the time, there will no longer be any good intentions. Though horrendous evil consequences would be completely absent, good intentions would also be completely absent. How does this compare to the actual world (with its mixture of both good and evil intentions and consequences)? A case could be made that the actual world, with a mixture of good and evil intentions and consequences, is morally better than worlds where good intentions are completely absent. After all, if this were the case, a HORRENDOUS-less world would have no morally good actions. Whatever else might be said about the actual world, at least it contains some morally good actions. Consequently, it's arguable that a HORRENDOUS-less world, despite having inhabitants with equivalent inner moralities, is indeed morally worse than the actual world.

It seems, therefore, that there is a way to maintain a Chen-style objection to Sterba's argument. Not only is it logically possible that all people would choose to act wrongly all the time in a HORRENDOUS-less world, there is a reason to believe that such a world is morally worse than the actual world, even if the 'inner moralities' of the people are the same across worlds. However, even if we grant that a HORRENDOUS-less world is morally worse than the actual world, does this show that Sterba's argument fails? I am not sure it does. Instead, what this may suggest is that there is a faulty assumption at the heart of the Chen–Sterba debate. The assumption is that God would actualize the best morally possible world (where the best morally possible world is assessed in terms of the quantity and distribution of good and evil intentions and consequences). What if the best morally possible world is simply not worth actualizing? Perhaps the lesson from this brief

discussion is *not* that God could have actualized a morally better world than the actual world as Sterba argues. It may turn out that a HORRENDOUS-less world is not morally better than the actual world. The lesson is that even if the actual world is the best morally possible world that could be actualized, God would nevertheless not actualize it because it violates MEPR1 *tout court*. Since the actual world exists, it follows that God does not exist, and a Sterba-style argument for God's non-existence remains more or less intact.

## 4. Conclusions

Hasker (2004, 2020, 2021) and Chen (2021) have raised objections against Sterba's argument for the non-existence of God based on the existence of horrendous evil consequences. Hasker argues that a HORRENDOUS-less world results in a moral kindergarten, while Sterba argues that it does not. Progress may be made in the debate between Hasker and Sterba by focusing on the *actual* way ordinary people engage in soul-making. Given that the existence of horrendous evil consequences in the actual world has little to do with how ordinary people conduct their lives, if we assume that ordinary people engage in soul-making, then it follows that a HORRENDOUS-less world does not result in a moral kindergarten.

Chen argues that it's logically possible that all people in a HORRENDOUS-less world choose to act wrongly all the time. She goes on to argue that such a world would be morally worse than the actual world. Sterba responds by showing that the 'inner moralities' of these people would be equivalent across worlds and that, all else being equal, the presence of horrendous evil consequences in the actual world makes it morally worse than the HORRENDOUS-less world under consideration. Though Sterba is right about the equivalence of inner moralities across worlds, all else is not equal because the presence or absence of evil intentions matters, and a case can be made for the moral inferiority of this HORRENDOUS-less world with respect to the actual world. Even if this were the case, it does not follow that Sterba's argument fails because it may still be argued that God should simply not actualize a world (even if it's the best morally possible world) because it violates MEPR1.

The upshot of this brief discussion is that it seems Sterba's argument has the potential to survive both Hasker's and Chen's criticisms.

**Funding:** This research received no external funding.

**Data Availability Statement:** Not applicable.

**Acknowledgments:** Many thanks to Rae Kim for providing access to bibliographic resources.

**Conflicts of Interest:** The author declares no conflict of interest.

## Note

[1] Thanks to an anonymous reviewer for raising this worry.

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
