# Peer review of "Limited Intervention and Moral Kindergartens"

_religions, doi:10.3390/rel13080729_

Round 1
Reviewer 1 Report
This is a fine paper. I do think it has some issues and room for improvement, though.
I confess from the outset that I'm not as familiar with the Sterba/Chen debate. That may be a happy situation, though, since I had the opportunity to read that section "from the outside," as it were.
At any rate, here are a few suggestions/thoughts:
First, there are a few places where I think the essay could be clearer. For example, you mention that Hasker refers to "the kinds of evils Sterba is concerned with" as "no gratuitous evil or NGE." Then, in Hasker's quote, you equate NGE with the principle of MEPR1. This second association seems more accurate to me. That is, NGE seems to refer to the principle of God being bound to facilitate a world without horrendous evil rather referring to the type of evil itself.
Another (albeit more nitpicky) example is when you say that Hasker is not alone in "defending" MKR. I'm not convinced "defending" is the best word here, but this issue is less pressing to me.
Second, I think you do a good job highlighting the impasse of Sterba and Hasker in terms of what is possible vis-a-vis soul-making in "moral kindergarten." However, I have at least one pressing question about your proposed categories as a solution. Your categories seem to suggest that an evil consequence must be objectively quantifiable. That is, an evil consequence either permits soul-making for everyone or is permits it for no one. This strikes me as an odd assumption, as people experience consequences very differently depending on a great many factors (including their personal history). As one example, a trauma victim may experience a later event/consequence as deeply troubling whereas another person may experience the same event/consequence as "no big deal." As another example--one that doesn't necessarily include past trauma--some people experience the death of a pet as devastating while others don't really feel much. It's unclear to me how your categories would accommodate this issue. (Perhaps you might say that there reaction to the event is the "consequence," which might work, but that seems to grind against the examples you use in your paper, which focus more on objective occurrences of torture and abuse than subjective experiences of those occurrences).
Third, I thought your criticism of Hasker's MKR (where you noted that most of us don't experience horrendous evils) was very strong. Well done there. (And I found the personal anecdote helpful). But I also wonder if Hasker might respond by saying that our *awareness* of the existence of horrendous evils allows for soul-building that would not be possible if such evils didn't occur. In other words, while I've never had the opportunity to free slaves, my awareness of the institution of slavery has impacted my moral development. Furthermore, if our world is, at least in part, profoundly shaped by horrendous evils (e.g., slavery and genocide), can we really say that we only live "in the residue" (as we would in a moral kindergarten in which such evils never occurred)? I'm not sure. Just a thought.
Fourth, my main issue comes with the section on the Chen-Sterba debate. I have to confess that I found this section a bit unclear and incomplete. Two issues surface here. First, there seems to be casual switched between discussing what was logically possible (what may be possible) and what actually was the case. Second, the discussion on evil intensions is never (so far as I can tell) accompanied by an explanation as to *why* there would be more evil intentions in a HORRENDOUS-less world. I acknowledge that this *may* be the case (it's logically possible), but I'm not seeing an argument that it *must* be the case. You note the Chen makes the argument, but you provide her support for it.
But then, you say that it's arguable that *all* possible HORRENDOUS-less worlds *are* worse that the actual world. This argument is based on the possibility of a HORRENDOUS-less world with all evil intentions (and thus no good intensions). But again, aside from this being a logical possibility, there's no rationale provided as to why such would be the case. For this reason, I take issue with the world "arguable." Aside from noting the logical possibility without any rationale supporting probability, it seems like you've gone a long way just to restate Chen's point: It's logically possible a HORRENDOUS-less is worse. Full stop.
(I may be missing something here. But I wonder if a revision of this section might help with the clarity).
Finally, I think breaking the conclusion away from the Sterba-Chen section and wrapping up the whole essay might be preferable. At the very least, I'd suggest a new paragraph starting at, "I don't think so." But I think a final paragraph summarizing how Sterba's argument seems to have the potential to survive both Hasker and Chen is worth adding.
All in all, a strong paper. I hope you find my comments are well-balanced on the annoying/helpful scale of reviews.
Author Response
Please see the attachment. My responses are in RED.

Reviewer 2 Report
There is not an extensive amount of research, but that is fine in this case because the article is a brief response to another scholar's argument. The response is concise and logically argued.
Author Response

(The authors gave the same response as above.)

Reviewer 3 Report
The most substantial notes I have for revision have to do with the style, the sources and structure of this paper.
First, the author begins with a clear analysis on the main topic. There is logical coherency and the argument is developed substantially. However, much work is required when it comes to the references used in order to built up an argument. More precisely, the author uses limited sources and it gives us the impression that ther is not enough engagement with additional material that could further substantiate and deepen the critical approaches offered here.
Next to that, there are not enough signposts. The author moves from one section to the other without providing sufficient explanation in regards to what the following section will be. I suggest that one or two sentences in the beginning of each section will be crucial for the reader to get on with the flow of the argument and a few sentences in the end of each section in order to summarize findings will contribute even better.
In addition, there are sentences that require citation. For example, 'one could follow Kantian intuitions and argue that evil intentions are morally worse than evil consequences' (p.5). It would be important for the author(s) to mention which particular work of Kant he/she/they are refereing to and to provide further explanation what this particular 'Kantian intuition' is about.
In general terms, this is a well written paper. However, the author must engage further with scholarship, reformulate the structure, ensuring at the same time that most claims are backed by evidence and most references to sources are accurately cited.
Author Response

(The authors gave the same response as above.)

Round 2
Reviewer 3 Report
The author has made substantial revisions to the manuscript. I remain impressed by the author's knowledge and I still think that the paper ought to be published. The additions in p.4 seem to be important for the text, as they make the less categorical and more balanced. The conclusion is re-written. The new version seems clearer and summarizes the findings in a much more precise way.
However, I think that the paper needs further attention on some issues. Consider, for example, in p.5 the following sentence in the footnote: 'An anonymous reviewer raised a potential worry that our awareness of the existence of horrendous evils...'. This, of course, adds important information to the topic. However, this claim should be referenced. Which particular source does make reference to this anonymous reviewer? Finally, it would be better for the author to integrate this sentence into the text.
As editor I feel that the manuscript certainly has merit. However, I am not sure about this part of the text (p.5). Claims uncited do not usually count as valid scientific claims. Even if the reviewer is anonymous the source must be mentioned. If this is not possible then it would be a good idea to search for the same argument somewhere else, in another work which could be accurately cited. Apart from this minor issue, the artice is in a good form.
